# Myriapod genomes reveal ancestral horizontal gene transfer and hormonal gene loss in millipedes

Wai Lok So[1,8], Wenyan Nong [1,8], Yichun Xie [1,8], Tobias Baril[2,8], Hai-yao Ma[3], Zhe Qu[1], Jasmine Haimovitz[4], Thomas Swale[4], Juan Diego Gaitan-Espitia [5], Kwok Fai Lau[6], Stephen S. Tobe [9], William G. Bendena [7], Zhen-peng Kai[3], Alexander Hayward [2✉] & Jerome H. L. Hui [1✉]

Animals display a fascinating diversity of body plans. Correspondingly, genomic analyses have revealed dynamic evolution of gene gains and losses among animal lineages. Here we sequence six new myriapod genomes (three millipedes, three centipedes) at key phylogenetic positions within this major but understudied arthropod lineage. We combine these with existing genomic resources to conduct a comparative analysis across all available myriapod genomes. We find that millipedes generally have considerably smaller genomes than centipedes, with the repeatome being a major contributor to genome size, driven by independent large gains of transposons in three centipede species. In contrast to millipedes, centipedes gained a large number of gene families after the subphyla diverged, with gains contributing to sensory and locomotory adaptations that facilitated their ecological shift to predation. We identify distinct horizontal gene transfer (HGT) events from bacteria to millipedes and centipedes, with no identifiable HGTs shared among all myriapods. Loss of *juvenile hormone O-methyltransferase*, a key enzyme in catalysing sesquiterpenoid hormone production in arthropods, was also revealed in all millipede lineages. Our findings suggest that the rapid evolution of distinct genomic pathways in centipede and millipede lineages following their divergence from the myriapod ancestor, was shaped by differing ecological pressures.

[1] School of Life Sciences, Simon F.S. Li Marine Science Laboratory, State Key Laboratory of Agrobiotechnology, The Chinese University of Hong Kong, Hong Kong, China. [2] University of Exeter, Exeter, UK. [3] School of Chemical and Environmental Engineering, Shanghai Institute of Technology, Shanghai, China. [4] Dovetail Genomics, Scotts Valley, CA, USA. [5] School of Biological Sciences, The Swire Institute of Marine Science, The University of Hong Kong, Hong Kong, China. [6] School of Life Sciences, The Chinese University of Hong Kong, Hong Kong, China. [7] Department of Biology, Queen's University, Kingston, ON, Canada. [8] These authors contributed equally: Wai Lok So, Wenyan Nong, Yichun Xie, Tobias Baril. [9] Deceased: Stephen S. Tobe. ✉email: alex.hayward@exeter.ac.uk; jeromehui@cuhk.edu.hk

The phylum Arthropoda contains the greatest diversity of animals, accounting for >80% of all described species. Myriapoda, including centipedes and millipedes, represents a diverse group of terrestrial arthropods containing ~16,000 described species, that play important ecological roles in soil and forest ecosystems. However, compared to many other arthropod lineages, the study of myriapod diversity and evolution has been relatively neglected[1,2].

Macroevolutionary trends in gene repertoire evolution have been analysed across many animal phyla, revealing that ecdysozoans, including arthropods, are characterised by remarkable patterns of gene loss[3,4]. Over recent years, the genomes of three myriapod species have been published: the geophilomorph centipede *Strigamia maritima*[1], the orange rosary millipede *Helicorthomorpha holstii* (Polydesmida) and rusty millipede *Trigoniulus corallinus* (Spirobolida)[2]. These genomic resources have provided important insights into gene family evolution in myriapods relative to other arthropods, yet, understanding of genome evolution between classes within this highly diverse group of animals remains extremely limited.

Several marine-to-terrestrial transitions of arthropod lineages occurred around 400 million years ago during the Silurian period, with myriapods representing one of the major terrestrial radiations[5]. Like other arthropods, myriapods possess a rigid chitinous exoskeleton that offers protection and prevents water loss. This exoskeleton is moulted periodically during ecdysis under the regulation of endocrine factors, including sesquiterpenoid hormones[6].

The subphylum Myriapoda comprises four classes of extant terrestrial arthropods: Chilopoda (centipedes), Diplopoda (millipedes), Pauropoda (pauropods) and Symphyla (symphylans/pseudocentipedes). Centipedes are further divided into five orders, and include >3000 species. In contrast to members of the other myriapod classes, centipedes are carnivorous, and possess unique venom injecting claw-like forcipules, located on the first trunk segment, which are used to incapacitate their prey. Millipedes are divided into sixteen orders and include >12,000 species[7]. Millipedes are detritivores that feed on leaf litter and act as important nutrient cyclers in terrestrial ecosystems. Pauropods and symphylans are also detritivores. Pauropods are divided into two orders and include ~830 species, while Symphyla contains just a single order of ~200 species.

In addition to their ecological relevance, myriapods occupy an important phylogenetic position in animal evolution, with analyses placing them as the extant sister group to Pancrustacea, the major animal clade containing insects and crustacean lineages[8]. Thus, consideration of the putative genomic character of the myriapod ancestor, and the subsequent divergent evolutionary pathways taken by centipede and millipede lineages, is of direct relevance to the study of arthropod evolutionary genomics. Yet, the current genomic resources available for myriapods are highly limited compared to other major arthropod lineages[9].

Here we seqeunce six new genomes, three centipedes and three millipedes, representing key unsampled lineages within Myriapoda, considerably increasing the genome resources available for this important group of arthropods. We then provide a comparative analysis of all available myriapod genomes (our six new genomes, plus three existing genomes[1,2]. Considerable variation in genome size is a distinctive feature of metazoan genomes, but the drivers of observed differences remain poorly understood[10]. Consequently, we start by investigating genome size across Myriapoda, to examine the extent of variation and potential causative factors. We then consider genomic features with potential roles in shaping the divergent evolutionary trajectories of centipedes and millipedes, towards active predation and detritivory, respectively. Specifically, we examine the extent to which patterns in gene family evolution (gene gain, loss and horizontal transfer) underlie key ecological and morphological differences. Such analyses offer fundamental insights into the mode of animal evolution[3], and provide an opportunity to assess whether genes that arose in other branches of life (such as fungi and bacteria), have driven the evolution of novel adaptations in animals following horizontal transfer[11,12]. We also compare the content and organisation of developmentally relevant genes and pathways between centipedes and millipedes, to examine their respective roles in myriapod evolution. In particular, we consider homeobox genes, the content and organisation of which can have profound impacts on animal phenotypes[13], and hormonal pathways which regulate arthropod development[6]. Collectively, our findings significantly expand current knowledge of the myriapod genomics, and provide novel insights into the evolution of genome size, gene repertoire, and genetic pathways across myriapod diversity.

## Results

**New myriapod genomes.** Using whole-genome sequencing strategies, we generated de novo genome assemblies for six myriapod species (Fig. 1A–G; Supplementary Data 1): centipedes *Lithobius niger* (Lithobiomorpha), *Rhysida immarginata* (Scolopendromorpha), *Thereuonema tuberculata* (Scutigeromorpha), and millipedes *Anaulaciulus tonginus* (Julidae), *Glomeris maerens* (Glomeridae), *Niponia nodulosa* (Cryptodesmidae) (Fig. 1A–G). Hi-C libraries were also constructed for the centipede *T. tuberculata* (Scutigeromorpha) and sequenced on the Illumina platform (Supplementary Data 1). The assembled myriapod genomes possess BUSCO scores of 63.4–93.8% (mean = 82.5%) (Fig. 1H). Orthologous gene families were used to reconstruct syntenic blocks between myriapod genomes. Scaffolds were ranked by orthologous gene family number. The one-to-one relationships of pseudo-chromosomal molecules or scaffolds were retained within centipedes or millipedes, as these can be identified and tested with strong significance. Overall, myriapod genomes display high genome synteny among species (Fig. 1H, Supplementary Figs. 1–3).

**Genome size evolution.** Genome size (i.e. assembly size) varies greatly among myriapods, with millipedes having much smaller genomes than centipedes, with the exception of *S. maritima* (Fig. 2A, Supplementary Data 2). Centipede genome size displays the greatest size variation, ranging greatly from just 176.2 Mb to 3.2 Gb, while variation in millipede genome size ranges from 149.0 Mb to 612.5 Mb (Fig. 2A, Supplementary Data 2). The size of the repeatome, genomic sequence comprised of transposable elements (TEs) and simple repeats, is known to correlate positively with genome size[14], and strong correlations between TE content and genome size have been observed for a variety of taxa, such as grasshoppers[15] ($r = 0.93$, $P = 0.0023$), Lepidoptera[16] ($r = 0.8$, $P = 0.001$) and fish[17] ($r = 0.47$, $P = 0.002$). Across all myriapod genomes, we identified an extremely strong positive relationship between TE content and genome size (Pearson's correlation test: $r = 0.98$, $t_7 = 12.051$, $P = 6.179e{-}06$; regression line: $y = 130 + 2x$; Fig. 2B), suggesting that TEs are a major contributor to genome size evolution for the group.

Considering centipede phylogeny (Fig. 2A), observed patterns in TE content and genome size suggest there was either: (i) a massive increase in TE content in the ancestral centipede lineage, followed by a similar reduction in TE content in *S. maritima*; or, (ii) parallel independent gains in TE content in *R. immarginata*, *L. niger* and *T. tuberculata*. Myriapod repeat landscape plots indicate that TE activity peaked relatively recently in myriapod genomes (i.e. most TEs are separated by low levels of genetic

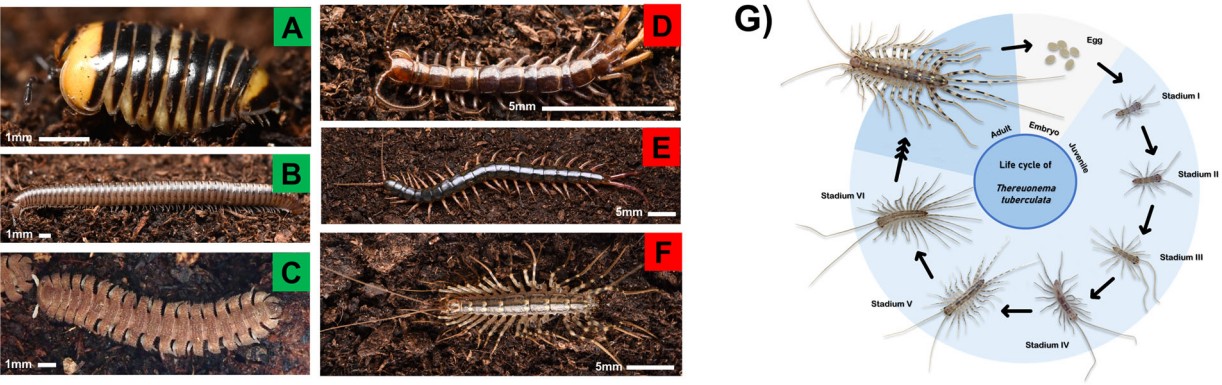

**Fig. 1 Myriapods used and sequenced in this study. A** *Glomeris maerens*; **B** *Anaulaciulus tonginus*; **C** *Niponia nodulosa*; **D** *Lithobius niger*; **E** *Rhysida immarginata*; **F** *Thereuonema tuberculata*; **G** Life cycle of *T. tuberculata*. **H** Genome sequencing information and statistics. The letters labelled in green are millipede species while the letters labelled in red are centipede species.

| | Species | Accession number | Assembly (bp) | N50 | BUSCO | Number of Proteins | Sum of Amino Acids | Mean of Proteins | Sum of Exons (bp) | Mean of Exons (bp) | Sum of Introns (bp) | Mean of Introns | Numer of gene loci | Sum of gene region (bp) | % of gene loci in genome | Average gene region (bp) |
|---|---|---|---|---|---|---|---|---|---|---|---|---|---|---|---|---|
| A | *Glomeris maerens* | WWPM00000000 | 149,886,780 | 10,323 | 81.50% | 23,034 | 8,485,375 | 368 | 33,452,515 | 239 | 33,494,403 | 288 | 20,807 | 56,062,638 | 37% | 2,694 |
| B | *Anaulaciulus tonginus* | WWPL00000000 | 612,114,675 | 23,611 | 87.00% | 47,951 | 18,244,321 | 380 | 63,549,906 | 351 | 109,220,826 | 828 | 47,482 | 159,845,628 | 26% | 3,366 |
| C | *Niponia nodulosa* | JAAIVG000000000 | 327,998,618 | 14,485 | 80.40% | 68,066 | 26,894,382 | 395 | 88,848,836 | 382 | 49,426,570 | 301 | 66,155 | 127,034,074 | 39% | 1,920 |
| | *Helicorthomorpha holstii* | JAAFCE000000000 | 181,195,123 | 18,119,263 | 91.80% | 22,989 | 10,831,902 | 471 | 41,851,668 | 250 | 77,481,553 | 537 | 20,787 | 100,135,091 | 55% | 4,817 |
| | *Trigoniulus corallinus* | JAAFCE000000000 | 448,518,024 | 26,787,286 | 85.90% | 20,761 | 9,411,701 | 453 | 40,130,240 | 244 | 210,057,821 | 1,470 | 19,072 | 196,236,880 | 44% | 10,289 |
| D | *Lithobius niger* | JAHWFP000000000 | 3,238,644,230 | 34,641 | 63.40% | 44,159 | 12,897,802 | 292 | 47,394,932 | 277 | 472,833,891 | 3,720 | 43,205 | 491,761,804 | 15% | 11,382 |
| E | *Rhysida immarginata* | JAHWFO000000000 | 2,528,959,310 | 956,869 | 78.60% | 29,419 | 10,747,316 | 365 | 41,355,813 | 239 | 677,140,419 | 4,748 | 28,320 | 609,236,451 | 24% | 21,513 |
| F | *Thereuonema tuberculata* | JAFIDM000000000 | 2,458,313,314 | 70,761,429 | 80.30% | 28,532 | 9,974,287 | 350 | 35,643,090 | 203 | 654,265,919 | 4,635 | 32,757 | 618,736,283 | 25% | 18,889 |
| | *Strigamia maritima* | GCA_000239455.1 | 176,208,967 | 139,451 | 93.80% | 15,008 | 7,227,006 | 482 | 32,765,509 | 247 | 40,934,796 | 350 | 14,992 | 73,239,189 | 42% | 4,885 |

distance), and expansion profiles of elements differ among centipede species (Supplementary Fig. 60). The main contributor to recent TE-related genome expansion in *R. immarginata* and *T. tuberculata* appears to have been DNA elements, while several TE groups have contributed to the particularly large expansion of genome size in *L. niger* (Supplementary Fig. 60). Conversely, *S. maritima* has experienced a more modest gain in genome size involving the expansion of long terminal repeat (LTR) TEs (Supplementary Fig. 60). To further examine the evidence for a decline in TE content in *S. maritima* versus independent gains in the other three centipede lineages, we calculated the number of shared vs unique TE families among sampled myriapod species. The results demonstrated that the overwhelming proportion of TE families are unique to individual species among currently sampled myriapod diversity (Fig. 2C, D; Supplementary Fig. 64). Thus, the data strongly support a hypothesis of parallel independent gains of TE content in *R. immarginata*, *L. niger* and *T. tuberculata*, rather than a reduction in TE content in *S. maritima*. More generally, these findings support the independent gain of new TE families as the main factor in shaping TE diversity across major divisions of myriapod diversity (Fig. 2A). This pattern is perhaps unsurprising considering the ancient divergence dates estimated to separate sampled major myriapod lineages[7,18] (e.g. estimated date of divergence between *Glomeris* and *Helicorthomorpha* is 456 MYA).

It is unclear why *S. maritima* has such a different TE profile compared to the other centipedes. *S. maritima* has a TE content of 69 Mb with a large LTR TE component, versus a TE content of 814 Mb in *T. tuberculata*, 1291 Mb in *R. immarginata* and 1610 Mb in *L. niger*, which all have much larger contributions from DNA TEs, long interspersed nuclear elements (LINEs) and rolling circle elements (Fig. 2A, Supplementary Data 2). A key ecological difference for *S. maritima* is that it is a coastal species that lives on the seashore, feeding on small marine molluscs, annelids, and crustaceans[19], in contrast to the other centipede species considered here, which are forest species that feed on terrestrial invertebrates. Over recent years it has become clear that horizontal transfer of transposons (HTT) is an important process in TE spread[20,21]. Thus, as voracious generalist predators, centipedes may be especially prone to accumulating TEs from their prey via HTT, with differences in diet potentially explaining differences in TE load between *S. maritima* and other centipedes, and also between centipedes and millipedes, which are detritivores that feed on leaf litter (Fig. 2A). However, the correlates of variation in TE content remain poorly elucidated, and we acknowledge that other life history traits may underlie observed patterns in TE abundance and diversity among myriapods. Effective population size has been suggested as a major potential driver of variation in TE content and genome size[14]. However, the suggested negative correlation is controversial, as it disappears when shared ancestry is taken into account using comparative phylogenetic approaches[22]. Further discussion is provided by Whitney[23], who state that among other issues, effective population size is often highly correlated with many other life history traits, and its relative importance remains far from clear.

TE load only partially explains differences in genome size observed among myriapods. The mean non-repetitive component of the genome is 214.8 Mb in millipedes, while it is more than four times greater in centipedes at 1154.7 Mb (Supplementary Data 2). Thus, other components of the genome appear to have also undergone expansion. We find that intron size in the three largest centipede genomes (*L. niger*, *R. immarginata* and *T. tuberculata*) is much greater than in millipedes (~3.7 kb to 4.7 kb per gene in centipedes, compared to 288 bp to 1.4 kb per gene in millipedes) (Fig. 1H). In sum, intron size contributes between 472.8 Mb and 677.1 Mb in the three centipedes, compared to 33.4 and 210.0 Mb in the five millipedes (Fig. 1H). Thus, expansion in intron size is also a major factor contributing to the large genome sizes of the centipedes *L. niger*, *R. immarginata* and *T. tuberculata*. Yet, considerable variation in myriapod genome

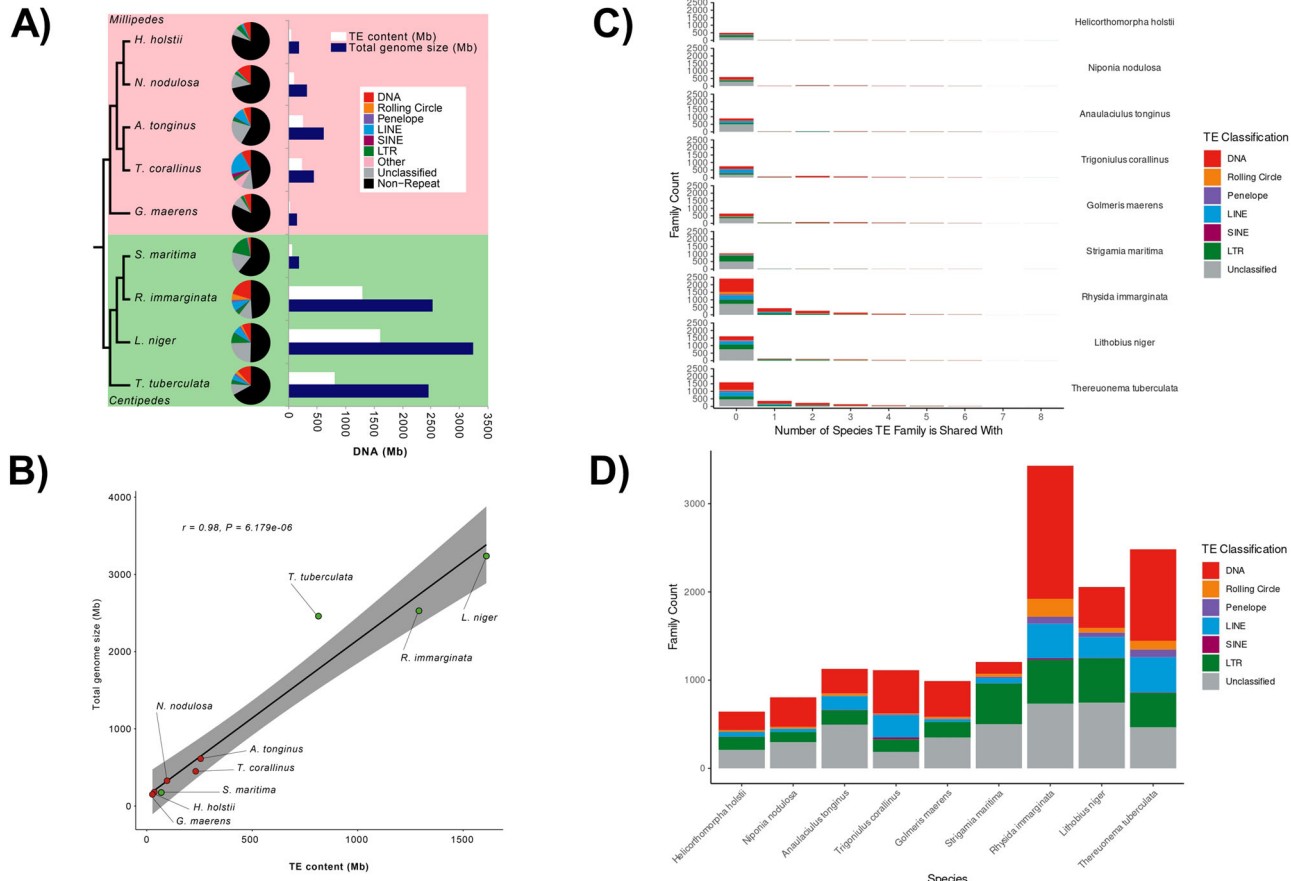

**Fig. 2 Transposable elements (TEs) analyses in myriapods. A** Summary of TE content in each myriapod genome. **B** Linear model predicting genome size based on TE content, with the grey area indicating the 95% confidence interval for predictions using the model. **C** Bar chart showing TE families that are shared among sampled myriapods. **D** Bar chart showing the diversity of TE families in each myriapod genome.

size remains even after accounting for repeats, introns and exons (Supplementary information – Supplementary Discussion).

**Gene gains**. To understand patterns in gene gain among myriapods, we first compared predicted genes among all myriapod genomes. Gene annotation was conducted using proteins from the uniprot_sprot database and transcriptomic data (Supplementary Data 1). The number of predicted protein-coding genes ranges from 20,761 to 68,066 for millipedes, and 15,008 to 44,159 for centipedes, accounting for 33.4 Mb to 88.8 Mb in millipede genomes, and 32.7 Mb to 47.3 Mb in centipede genomes (Fig. 1H). We identified relatively few gains in gene families in the last common ancestor of millipedes compared to the last common ancestor of centipedes, which gained a large number of new gene families after the two subphyla diverged (Fig. 3A, B). We then tested if any functional gene ontology terms, including these from GO, KEGG and KOG databases (Supplementary Figs. 4–7, Supplementary Data 4–6), were enriched in either the millipede ancestor or centipede ancestor (i.e. shared by all 5 millipedes, but absent in centipede genomes, or vice versa). As shown in Fig. 3C and the Supplementary Data 4–6, seven KEGG pathways were enriched in the centipede ancestor, including the adipocytokine signalling pathway, complement and coagulation cascades, isoquinoline alkaloid biosynthesis pathway, PPAR signalling pathway, SNARE interactions in vesicular transport, fatty acid biosynthesis and tyrosine metabolism. Considering that the enrichment of orthologues in several of these pathways involves lipid metabolism, immunity and signal transduction, these may

also have contributed to sensory and locomotory adaptations that facilitated the ecological shift to predation in centipedes.

**Homeobox genes**. Homeobox genes, including ParaHox and Hox genes, are crucial transcription factors to understand animal evolution[13,24]. Comparison across myriapod genomes shows high conservation in the Hox and ParaHox genes (Fig. 4A, B, Supplementary Figs. 8–9, Supplementary Data 7). In addition, loss of the ParaHox gene *Pdx/Xlox* was apparent across all myriapod genomes, as documented in previous studies considering fewer taxa[1,2] (Fig. 4A, B). Authors previously suggested genomic "relaxation" of the position of *Hox3*, resulting in its translocation outside the Hox cluster in the myriapod ancestor[1,2]. In the newly sequenced centipede genomes of *T. tuberculata* and *R. immarginata*, an intact Hox cluster (*Lab/Pb/Hox3/Dfd/Scr/Ftz/Antp/Ubx/AbdA/AbdB*) was identified (Fig. 4A, Supplementary Fig. 8). In the other myriapod genomes investigated here, Hox genes are scattered across different scaffolds. However, the organisation of the Hox cluster genes identified in the new myriapod genomes contradicts the previous hypothesis, instead suggesting that an intact Hox cluster was present in the myriapod ancestor, with subsequent loss of functional constraints for *Hox3* (or possible technical assembly problems) in certain myriapod lineages only (Fig. 4A).

**Horizontal gene transfer in centipede ancestor**. Horizontal gene transfer (HGT) between lineages has contributed to the evolution of novel adaptive traits among animal diversity (Fig. 5A). In a recent study utilising centipede venom gland transcriptome data,

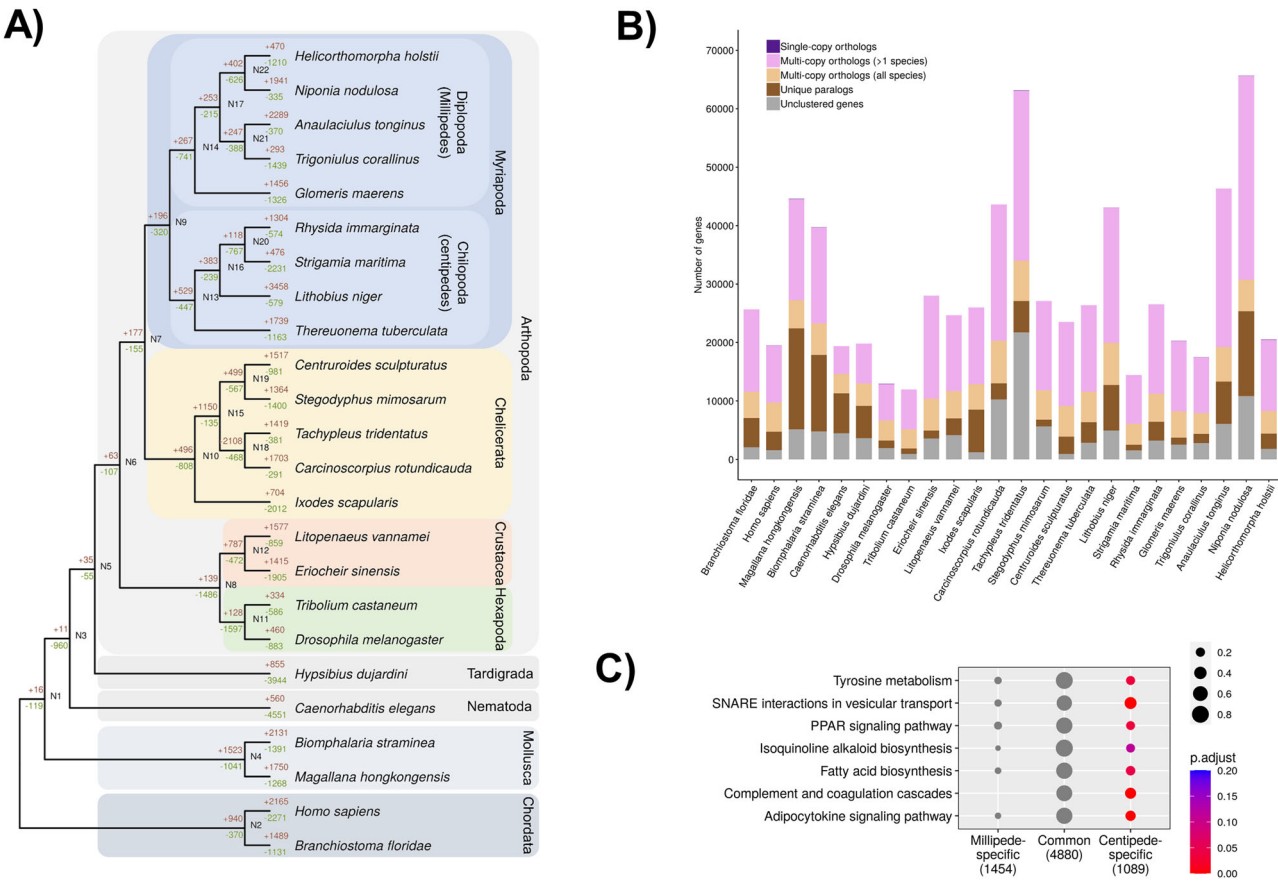

**Fig. 3 Gene gain and loss in myriapod genomes. A** Summary of gene gain and loss in myriapods and other reference animals. **B** Number of shared ortholog groups in myriapods and other reference animals. **C** Seven enriched KEGG pathways of gene gains in centipedes. *P* value is calculated based on the hypergeometric model and *P* value is adjusted using Benjamini and Hochberg method (BH).

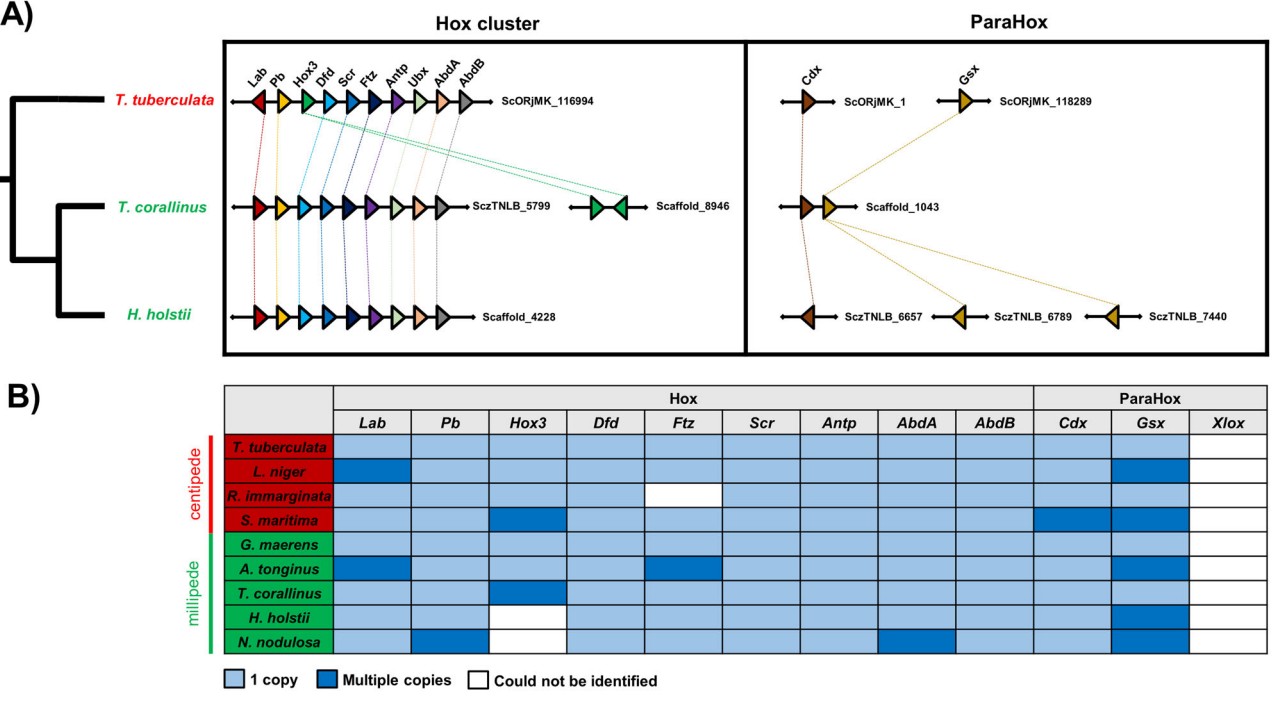

**Fig. 4 Hox and ParaHox in sequenced myriapod genomes. A** Microsynteny of Hox and ParaHox clusters in *H. holstii*, *T. corallinus* and *T. tuberculata*. **B** Hox and ParaHox genes annotation in sequenced myriapod genomes. The species labelled in green are millipede species while the species labelled in red are centipede species.

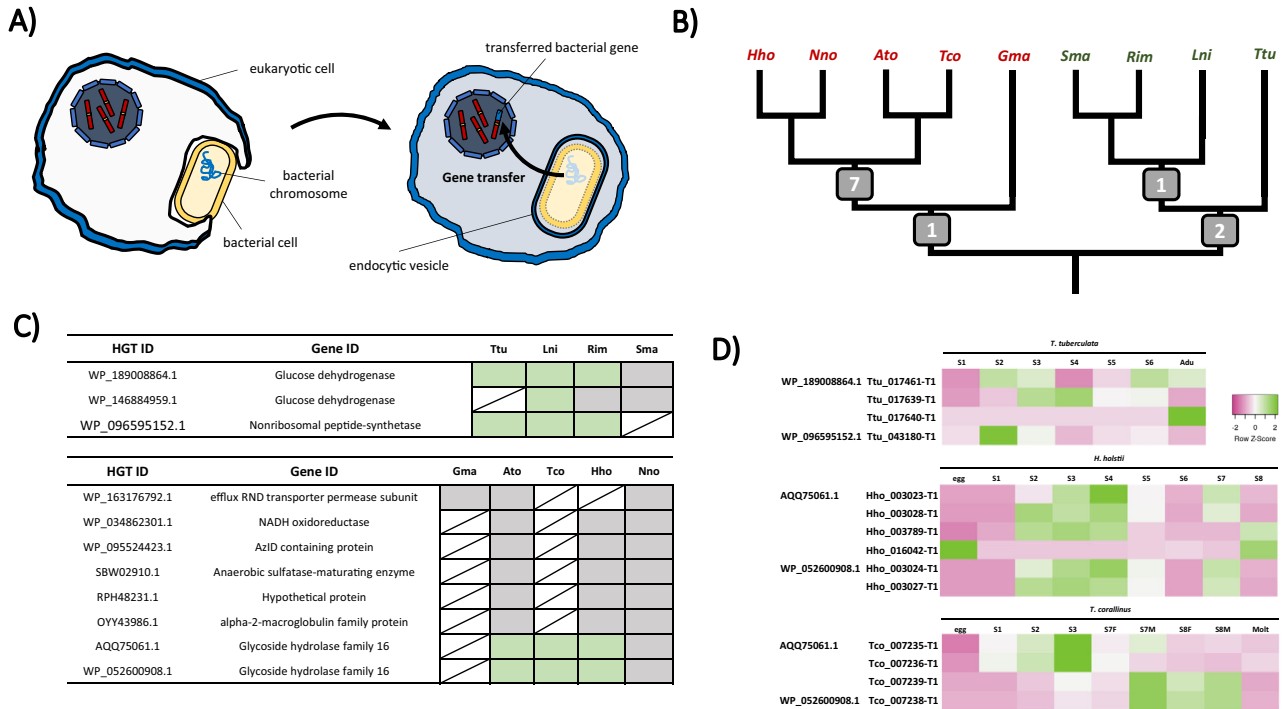

**Fig. 5 Horizontal gene transfer (HGT) in myriapods. A** Schematic diagram showing a typical HGT events from prokaryotic cells to eukaryotic cell. **B** Summary of HGT events identified in different myriapod lineage in this study. The numbers at the nodes indicate the HGT event identified in each lineage common ancestor. **C** HGT candidates identified and their presence (labelled in green) in each myriapod lineage. **D** Gene expression of HGT candidates in *T. tuberculata*, H. holstii and *T. corallinus* transcriptomes (this study and Qu et al., 2020) The colours indicate the degree of row Z score (green indicates upregulation while pink indicates downregulation).

it was demonstrated that centipede genomes have recruited gene families from bacterial and fungal donors during venom evolution[25]. By obtaining the most comprehensive list of myriapod genomes yet assembled, here we ask what is the pattern of HGT events among myriapods?

Detection of potential HGT candidates in each myriapod genome was first annotated by the programme "Alienness", which predicts targets through a BLAST search approach using the NCBI public database. The putative HGT dataset was then examined through a microsyntenic approach to search for conserved loci patterns among myriapod genomes, excluding genes that locate on a short scaffold without flanking genes. Conserved syntenic patterns found across at least three species were regarded as confident HGT candidates and resultant candidates were further examined by phylogenetic analyses and consideration of gene structure.

Using the above approach, we identified 11 cases of HGT in myriapod genomes (Fig. 5B, C, Supplementary Figs. 10–29, Supplementary Data 8–18). Among the resultant 65 genes across the 9 myriapod genomes identified as originating from these 11 cases of HGT, 32 do not have introns and 33 possess introns. A total of 8 transferred genes were discovered in millipedes (Fig. 5B), with 1 case (*efflux RND transporter permease subunit*) shared by all sequenced genomes and the remaining 7 (*NADH oxidoreductase*, *AzlD domain-containing protein*, *anaerobic sulfatase maturase*, *alpha-2-macroglobulin*, *SYLF domain-containing protein* and two *glycoside hydrolase family 16 proteins* (*GH16*)) recognized in the genomes *T. corallinus*, *A. tonginus*, *N. nodulosa* and *H. holstii* (the helminthomorphs). While 3 cases were recognized in the centipede lineage (Fig. 5B), with 2 transferred genes (*glucose dehydrogenase* (*GDH*) and *non-ribosomal protein synthetase*) present in the centipede common ancestor and 1 (another *GDH*) shared by *L. niger*, *R. immarginata* and *S.*

*maritima* (the pleurostigmomorphs). All identified cases of HGT to myriapods originated from bacterial genomes, as confirmed by phylogenetic analyses and matching best hit results of the BLAST search on NCBI database, illustrating the bacterial origin of the identified genes.

Among the transferred genes in millipedes, gene expression was only observed for the *GH16*s (Fig. 5D). Both genes display expression in *A. tonignus*, *T. corallinus* and *H. hostii*, with a relatively greater amount of expression during the early post-embryonic developmental stages and the pre-adult stage of *T. corallinus* and *H. holstii* (Fig. 5D). GH16 consists of a family of glycosidases and transglycosidases found in both prokaryotes and eukaryotes, which hydrolyse the β-1,3(4)-linkages of natural polysaccharides. Various types of glucoside hydrolases are present in the genomes of phytophagous insects including beetles and moths[26–28]. While in the nematode *Aphelenchoides besseyi* and *Aedes aegypti* mosquito larvae, GH16 is suggested to be associated with glucan rich diets[29,30]. Previous studies on centipedes have also illustrated the presence of GH in venoms, where they are postulated to act as "spreading factors" that promote the pathological effects of other venom elements after envenomation, and also as antimicrobial factors[31]. However, clarification of whether the GH16s identified in millipedes are involved in nutrition or defence requires further investigation.

In centipedes, all transferred genes were found to be expressed, except for in *S. maritima* (Fig. 5C). Non-ribosomal protein synthetase (NRPSs) are large multifunctional enzymes that synthesize small bioactive peptides in bacteria and fungi, which contribute to formation of siderophores, pigments and toxins[32,33]. Thus, they are proposed to have been contributed by HGT between prokaryotes and eukaryotes[32,34]. They have also been found in other animals, for example high structural resemblance between NRPSs and the pigmentation gene Ebony in the fruitfly *Drosophila*

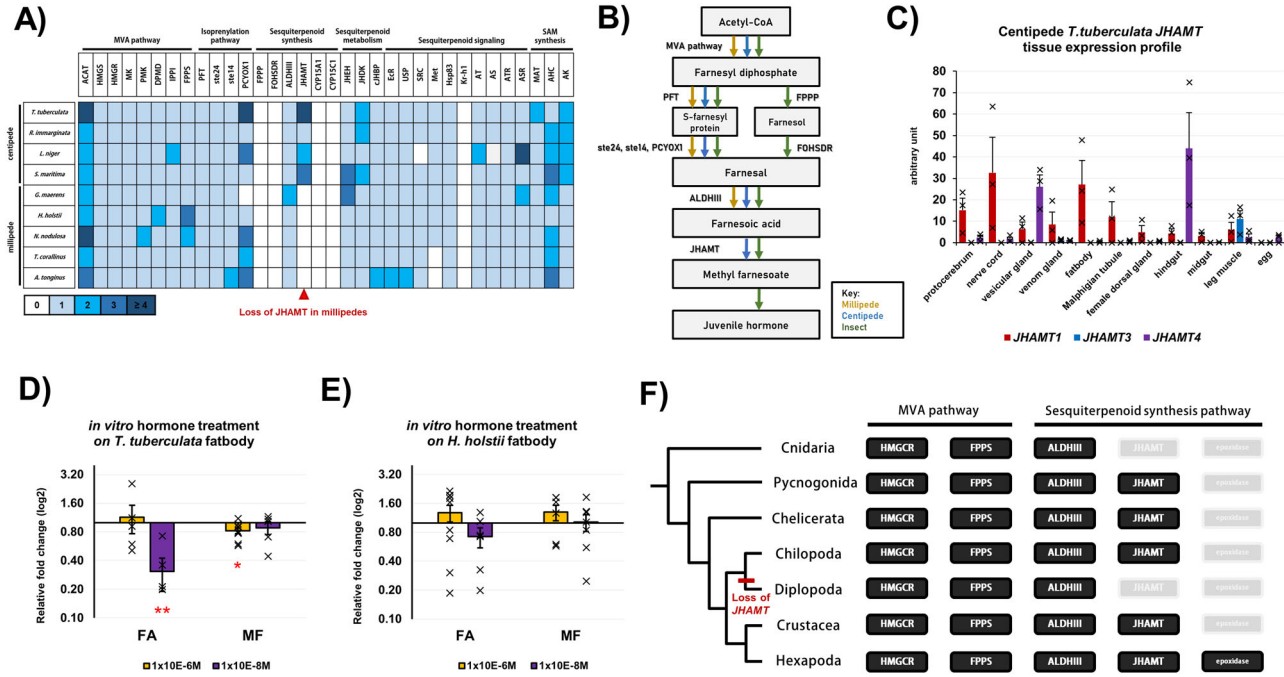

**Fig. 6 Sesquiterpenoid system in myriapods. A** Sesquiterpenoid pathway genes identified in sequenced myriapod genomes. **B** Summary diagram indicating different pathways that arthropod utilized for the synthesis of different sesquiterpenoid hormone. **C** The differential tissue expression of *JHAMT* in *T. tuberculata* ($n = 3$ biologically independent *T. tuberculata*). Data are presented as mean values $+/-$ SEM. Individual data points are present on each bar. The vertical error bars indicate the standard error of mean (SEM) among the data points in each experiment. **D** The *vitellogenin* (*Vg*) expression in *T. tuberculata* fatbody ($n = 5$ biologically independent *T. tuberculata* fatbody tissue for $10^{-6}$ M FA in vitro treatment, $n = 5$ for $10^{-8}$ M FA treatment, $n = 9$ for $10^{-6}$ M MF treatment, $n = 5$ for $10^{-8}$ M MF treatment) post-hormonal treatment for 4 h. Data are presented as mean values $+/-$ SEM. Individual data points are present on each bar. The vertical error bars indicate the standard error of mean (SEM) among the data points in each experiment. Two-tailed student *t*-test with two sample unequal variance test was conducted with confidence interval at 0.95. Red asterisk indicates the *P* value of the test, **$\leq$0.05 and *$\leq$0.08. The *P* value of $10^{-8}$ M FA experiment is 0.002915, while that of $10^{-6}$ M MF experiment is 0.07234 (as shown in the figure). **E** The *Vg* expression in *H. holstii* fatbody ($n = 9$ biologically independent *H. holstii* fatbody tissue for $10^{-6}$ M FA in vitro treatment, $n = 6$ for $10^{-8}$ M FA treatment, $n = 6$ for $10^{-6}$ M MF treatment, $n = 7$ for $10^{-8}$ M MF treatment) post-hormonal treatment for 4 h. Data are presented as mean values $+/-$ SEM. Individual data points are present on each bar. The vertical error bars indicate the standard error of mean (SEM) among the data points in each experiment. Two-tailed student *t*-test with two sample unequal variance test was conducted with confidence interval at 0.95. **F** Summary indicating the evolution of sesquiterpenoid pathways in arthropods and cnidarians.

*melanogaster* has also been reported[35]. The two *GDHs* identified here resemble *PQQ-sugar dehydrogenase*, which the protein uses pyrroloquinoline-quinone (PQQ) as cofactor to oxidise glucose to gluconolactone[36]. PQQ-dependent enzymes promote the oxidation of alcohols and sugars in the periplasmic space of gram-negative bacteria, but there is little evidence that eukaryotes use PQQs as cofactors, except in fungi[37–40]. Nevertheless, the presence of gluconolactone has been recognized in arthropods, including the mite *Tetranychus urticae*, and in the defence secretions of cockroaches of *Eurycotis spp.*[41–43]. A temporal expression of GDH was observed in the transcriptomes of *T. tuberculata*, and the expression of *GDH* in different tissues of *T. tuberculata* was also examined using quantitative real-time PCR, revealing that in addition to the venom gland, *GDHs* are also highly expressed in the gut and leg (Supplementary Fig. 30).

**Sesquiterpenoid hormone loss in millipede ancestor.** Arthropod moulting, metamorphosis, and reproduction are precisely controlled by the interaction of two hormonal systems: ecdysteroids for cuticle moulting, and sesquiterpenoids for postembryonic growth and differentiation[44–46]. The sesquiterpenoid hormonal system is an ancient system present in the cnidarian–bilaterian ancestor, and the sesquiterpenoids farnesoic acid (FA), methyl farnesoate (MF), and juvenile hormone (JH) are produced through the conserved mevalonate (MVA) pathway in animals[6,47,48].

Among the myriapod genomes investigated here, all genes in the MVA pathway were identified (Fig. 6A, Supplementary Data 19–27, Supplementary Figs. 31–55). For the sesquiterpenoid-specific pathway genes, all were identified in the four centipedes, but the *juvenile hormone O-methyltransferase* (*JHAMT*) gene was missing in all five millipede genomes (Fig. 6A, B, Supplementary Data 28, Supplementary Figs. 56, 57). In insects and crustaceans, JHAMT is a key enzyme responsible for catalysing the last step of sesquiterpenoid hormone MF production, which regulates development and reproduction[44,48].

To address the significance of *JHAMT* loss in the millipede ancestor, we first tested whether JHAMT is functional in centipedes. As shown in Fig. 6C, expression of *JHAMTs* was identified in different tissues including the protocerebrum, nerve cord, vesicular gland, fat body, gut and legs. Moreover, hormone measurement was conducted, to reveal the FA and MF titre in *T. tuberculata* and *H. holstii* (Supplementary Fig. 58). In *T. tuberculata*, a high FA titre with a trace titre of MF was observed. While in *H. holstii*, the titre of both FA and MF remain low. In vitro treatment of the sesquiterpenoids FA and MF was then performed based on the measured physiological titre. In the fat body of the centipede *T. tuberculata*, the treatment significantly modulated the expression of *vitellogenin* (*Vg*), the precursor of egg yolk that is involved in reproduction in insects and crustaceans[49] (Fig. 6D). In contrast, we carried out an in vitro treatment of the sesquiterpenoids FA and MF in the fat body of

the millipede *H. holstii*, which resulted in no change in the expression of *Vg* (Fig. 6E).

Thus, our data demonstrate that centipedes have a functional sesquiterpenoid hormone system similar to that of insects, where the reproductive gene *Vg* is regulated by sesquiterpenoids[49,50]. Conversely, after the loss of *JHAMT* in the millipede ancestor, regulation of *Vg* by sesquiterpenoids has been lost (Fig. 6F). How this loss contributed to the different reproductive modes present among millipedes remains to be investigated.

## Discussion

With reference to newly sequenced centipede and millipede genomes, we compare diverse genomic features including genome size, transposable element (TE) content, genomic architecture (organisation of Hox genes), and evolutionary processes mediated by gene gains (horizontal gene transfer/HGT) versus gene losses (sesquiterpenoid hormone system). Our findings expand current understanding of the processes mediating genome evolution and macroevolutionary dynamics for Myriapoda, a diverse, abundant, and geographically widespread, but relatively neglected group of terrestrial arthropods.

The genomic complexity of metazoans is traditionally characterised by evolutionary dynamics that involve substantial changes in genome size and genome architecture. Genome size evolution is a fundamental biological characteristic, but the causes regulating it are poorly characterised[10]. Among myriapods, genome size was found to be highly variable. Centipedes generally have much larger genome sizes than millipedes, except for the geophilomorph centipede lineage (i.e. *Strigamia*), which has a comparatively small genome, consistent with other reports of small genome size for this group[51]. We show that expansions in intron size and TE content are important contributing factors to observed genome size differences in centipedes and millipedes, and demonstrate that an additional major contribution to genome size has occurred, through as of yet undetermined processes. Future attempts to address finer-scale mechanisms underlying myriapod genome size evolutionary dynamics (including genome turnover[52]) by considering whole genome alignments, will require greater taxon sampling than is currently available, given the great evolutionary distances that separate currently sampled taxa (e.g. the divergence between Diplopoda and Myriapoda is estimated to have occurred sometime from the mid-Silurian to the Cambrian[53]). Thus, we highlight the need for further genome sequencing to further understanding of this important group of terrestrial arthropods.

The order and content of highly conserved genes, such as homeobox genes, provides an important context to understand relationships among genotype and phenotype in animals. The new centipede genomes provided here reveal an intact Hox cluster containing all expected *Hox*. This suggests that the loss of functional constraints on *Hox3* observed in different myriapods occurred after the diversification of the myriapod ancestor. In myriapods, *Hox3* has been found to retain an ancestral Hox-like expression pattern[54–56], while in insects *Hox3* has duplicated copies in different lineages (i.e. *zen*/*bicoid*)[57]. The findings reported here expand the view that the *Hox3* family has undergone dynamic changes in different lineages of myriapods and other arthropods, although the evolutionary causes remain unclear.

The horizontal gain of genes from other lineages, also known as HGT, is considered an important process in animal adaptation, but is poorly examined in myriapods. We demonstrate several novel shared HGT events in the genomes of centipedes and millipedes, but not between the two lineages. Our findings suggest that HGT events occurred independently in the ancestor of

centipedes and the ancestor of millipedes after their separation from the myriapod ancestor, and were related to the divergent adaptive evolution of centipedes to the niche of venomous predators.

Loss of genes can also affect phenotypic evolution, but the evolutionary consequences are often more difficult to address. We show that a key enzyme in the sesquiterpenoid hormone system has been lost in all millipede lineages, resulting in the loss of functional regulation of a key reproductive gene. This finding suggests that gene loss events can occur in the endocrine system, contrary to the conventional view of stepwise gene gain being responsible for generating diversity in animal physiology. Further experiments are thus necessary to elucidate the implications of the loss of this purportedly key gene, and to test if alternative hormones regulate reproduction in millipedes.

More generally, the myriapod genomes presented here also provide insights into the putative genomic architecture of the myriapod ancestor, and the divergent pathways followed by centipede and millipede lineages following their divergence from this ancestor. This remarkable divergence has led to two very different lifestyles being expressed in extant myriapods: (i) predation in centipedes, characterised by the evolution of offensive chemical weaponry in the form of potent venoms, and (ii) detritivory in millipedes. We provide the first steps towards unravelling the genomic bases of the divergent adaptations underlying these two very different ecologies.

## Methods

**Ethical statement**. We have complied with all relevant ethical regulations for animal research.

**Animal collection and husbandry**. Adult *R. immarginata*, *L. niger* and *A. tonginus* (10 for each sex) were collected in an agricultural garden of New Asia College, The Chinese University of Hong Kong. Mixed sexes and different developmental stages of 30 adult *T. tuberculata*, 20 adult *G. maerens* and 15 adult *N. nodulosa* were purchased from an exotic pet shop in Hong Kong. Species identity was confirmed by DNA Sanger sequencing of the mitochondrial cytochrome oxidase subunit I (*COI*) gene, using the universal primers, LCO1490 (5′-GGTCAACAAATCA-TAAAGATATTGG-3′) and HCO2198 (5′-TAAACTTCAGGGTGAC-CAAAAAATCA-3′)[58]. Captured animals were kept in plastic tanks at room temperature. The tanks were filled with slightly moistened gardening soil, mixed with Zoo Med's Repti Calcium powder (Repti Calcium®, Zoo Med Laboratories, Inc, USA, A34-8), as a substratum. A dish of tap water and hydrated dried sphagnum moss was provided as a source of water. Dried leaves were collected and autoclaved and then transferred on top of the soil. Lettuce pieces and juvenile live crickets were provided for millipedes and centipedes, respectively, and tap water was sprayed in a 2 to 4-day interval to maintain soil humidity. Photographs were taken using an Olympus Tough TG-6 Digital Camera.

**Genome sequencing**. Genomic DNA (gDNA) was extracted from each adult species from one individual, using the PureLink Genomic DNA Mini Kit (Invitrogen™, Catalogue number: K182001) following the manufacturer's protocol.

Extracted gDNA was examined by gel electrophoresis to ensure clear intact bands for sequencing inputs. Qualified samples were delivered to Novogene, and Dovetail Genomics for initial library preparation and subsequent genome sequencing. Meanwhile, a Chicago library was also prepared by Dovetail Genomics with the preparation method described previously by Putnam[59]. In detail, around 500 ng of gDNA (mean fragment length = 55 kb) from the animal sample was artificially reconstituted into chromatin and fixed with formaldehyde solution. Fixed chromatin was then digested with restriction enzyme DpnII. The 5′ overhangs produced were lined with biotinylated nucleotides, and the free blunt ends were ligated afterwards. Crosslinks of molecules were then reversed, and the DNA molecules were purified from proteins. Purified DNA was treated to remove Biotins that were not internal to the ligated fragments were then removed from the purified DNA. The DNA molecules were then sheared to ~350 bp of mean fragment size. The sequencing libraries were created using NEBNext Ultra enzymes and Illumina-compatible adapters. Biotin-containing fragments were separated using streptavidin beads prior to PCR enrichment of each sequencing library. The libraries were then loaded and sequenced on the Illumina HiSeq X platform.

For the additional Dovetail Omni-C library of *T. tuberculata*, chromatins were fixed with formaldehyde and then extracted as described above. Fixed chromatins were digested with the restriction enzyme DNAse I and the chromatin ends were fixed and ligated to biotinylated bridge-adapter followed by subsequent proximity

ligation of the primer ends. Crosslinks were reversed, and the DNA molecules were purified afterwards. Excess biotins were removed from the purified DNA molecules. The sequencing library was created using NEBNext Ultra enzymes and Illumina-compatible adapters. Biotin-containing fragments were separated using streptavidin beads prior to PCR enrichment of each sequencing library. The library was then sequenced on the Illumina HiSeqX platform to produce 449 M read pairs.

**Transcriptome sequencing**. Transcriptomes of each species were sequenced at Novogene. Total RNA from different tissues was isolated using TRIzol reagent (Invitrogen™, Catalogue number: 15596026) according to the manufacturer's instructions, and subjected to quality control using a NanoDrop™ One/OneC Microvolume UV-Vis Spectrophotometer (Thermo Scientific™, Catalogue number: ND-ONE-W), gel electrophoresis, and Agilent 2100 Bioanalyzer (Agilent RNA 6000 Nano Kit).

Qualified RNA samples were proceeded to library preparation and sequencing at Novogene. PolyA-selected RNA-Sequencing libraries were constructed using the TruSeq RNA Sample Prep Kit v2. The concentrations and mean insert sizes of the final prepared libraries were assessed using quantitative real-time PCR (TaqMan Probe) and the Agilent 2100 bioanalyzer (Agilent DNA 1000 Reagents), respectively.

**Genome assembly**. Chromium WGS reads were checked and contamination was removed using Kraken2[60] and then separately used to make a de novo assembly using Supernova (v 2.1.1), with default parameters. The dedupe contigs of the pseudohap style output were used for the following analysis. For *T. tuberculata*, the de novo assembled and Dovetail Omni-C library reads were used as input data for genome scaffolding by HiRise[59]. Sequences from Dovetail Omni-C library were aligned to the draft assembly using the software bwa (https://github.com/lh3/bwa). The mapped reads were analysed in HiRise[59]. A likelihood model was produced for genomic distance between mapped pairs, which was subsequently used to locate, to break predicted misjoins, to score prospective joins, and to create joins above a threshold.

**Gene model prediction**. RNA-seq data were pre-processed with Trimmomatic[61] and converted to transcripts using genome-guided Trinity[62], AUGUSTUS[63] was trained using BUSCO[64], and GeneMark-ES[65] was used for ab initio gene prediction. Gene models were trained and predicted by funannotate[66] using the parameters "--repeats2evm --protein_evidence uniprot_sprot.fasta --genemark_mode ET --busco_seed_species fly --optimize_augustus --busco_db arthropoda --organism other --max_intronlen 350000", the gene models from several prediction sources including GeneMark, high-quality Augustus predictions (HiQ), pasa, Augustus, GlimmerHM and snap were passed to Evidence Modeler to generate the gene model annotation files. PASA was used to update the EVM consensus predictions, UTR annotations were added, and models for alternatively spliced isoforms were created. The protein-coding genes were searched with BLASTP against the nr and swissprot databases by diamond (v0.9.24)[67] with parameters "--more-sensitive --evalue 1e-3", and mapped by HISAT2 (version 2.1.0)[68] with transcriptome reads. Gene models with no similarity to any known protein and no mRNA support were removed from the final version.

**Identification of orthologous genes and gene families**. Orthologues and orthogroups in the proteomes of 9 myriapods and 24 reference outgroups were inferred using the software OrthoFinder v. 2.5.2[69] with default values setting and '-M msa' activated. The longest predicted protein of each individual gene was used as the representative inputs for the OrthoFinder analysis. Gene duplication ratios at each node/tip were computed by dividing the number of duplications observed in each node/tip by the total number of gene trees containing that corresponding node. The software CAFE5 was then applied to infer gene gain and loss rates in each genome[70]. The orthogroups generated by OrthoFinder were regarded as distinct gene families and provide as inputs for CAFE5 analysis. The divergence tree was inferred using r8s[71] from the species tree produced by OrthoFinder.

**Functional terms enrichment analysis**. The functional term annotations were performed using eggNOG[72]. Orthogroups were assigned with Gene Ontology (GO), EuKaryotic Orthologous Groups (KOG), Kyoto Encyclopedia of Genes and Genomes (KEGG) and KEGG Orthology (KO) terms by inheriting the terms from genes found within the groups. Functional enrich was tested using function 'compareCluster()' in R package 'clusterProfiler' v.3.16.1[73] under the environment of R 4.1.0[74]. Significantly enriched terms were determined with pvalueCutoff = 0.05, pAdjustMethod = "BH", and qvalueCutoff = 0.2. The genome of internal nodes (ancestral populations) was inferred according to the gene count results of CAFE5. Data were visualised using R packages 'ggplot2'[75], 'ggtree'[76] and 'pathview'[77].

**Macrosynteny analysis**. Orthologous genes were anchored by mutual best Diamond BLAST hits (--evalue 0.001). Synteny Oxford dot plots were generated by indexing and comparing the orthologue positions and numbers. Plotting order were determined by the length of pseudochromosomes (scaffolds) and number of

ortholog pairs. Significance was tested based on the one-tailed test for hypergeometric distribution using Fisher's exact test[78], and the p value was adjusted using "Benjamini-Hochberg" method. Results were visualised using R package 'ggplot2'[75].

**Repeat annotation**. Repetitive elements were identified using the Earl Grey TE annotation pipeline (version 1.2) (https://github.com/TobyBaril/EarlGrey)[79]. Earl Grey was run specifying the *arthropoda* (-r arthropoda) subsets of RepBase (version 23.08)[80] and Dfam (version 3.3)[81] to mask known repeats prior to the de novo TE annotation steps. Subsequently, Earl Grey identified and optimised de novo TEs using an automated implementation of the "BLAST, Extract, Extend" process[82]. Redundant consensus sequences were removed from the TE library before the de novo and known repeat libraries were combined for annotation of the genome assembly. Finally, Earl Grey processes the annotations to resolve overlaps and defragment repeat loci before final TE quantification. Quantification and subsequent analysis were performed in R (version 3.6.3), using the Rstudio IDE[74]. Plots were generated using ggplot2 of the tidyverse package[75,83]. For the chromosomal-level assemblies, karyoplots illustrating gene loci and TE density across chromosomes (*window.size = 10000*) were generated in R using the karyoploteR package[84].

**Identifying shared and unique TEs in the myriapods**. To classify TEs as shared or unique among sampled myriapods, de novo TE libraries were clustered using CD-Hit-Est (-d 0 -aS 0.8 -c 0.8 -G 0 -g 1 -b 500 -r 1), with TEs being defined as a family if similarity exceeded the thresholds described by Goubert[85]. Clusters were analysed in R[86,87] to assign shared vs unique status to TE family, whereby a share count of 0 defined a TE family unique to a single species, while a share count of 1–8 defined a TE family as shared across other myriapod genomes. Using the final TE annotations, shared status was also quantified for families present in the Dfam and RepBase libraries, and these were combined with the de novo family results. The results were plotted in R using GGplot2 from the tidyverse package[75,83], separating TE families according to shared states and by TE classification.

**In silico horizontal gene transfer (HGT) analyses**. Protein-coding genes from myriapod genomes were used as queries in BLASTp searches against the nr database in diamond (v0.9.24) specifying the parameter "--evalue 1e-3 --outfmt 6", after which they were submitted to Alienness[88] using the parameters described in Undheim and Jenner[25], to detect potential HGT candidates. Alienness calculates an index for each query sequence present in the genomes based on the *E*-values of the BLAST results, indicating the potential donor (non-metazoan) and recipient (metazoan) taxa. A conserved domain search was performed using the online NCBI Conserved Domain Search tool. Shared potential HGT candidates were subsequently investigated in detail by identifying microsyntenic patterns in the respective genomes. Microsynteny analysis was performed on the 5 genes located at up-stream or down-stream region of each HGT gene, respectively. For HGT candidates with less than 5 genes up/down-stream, all surrounding genes will be included in the analysis. Scaffolds contained only the HGT were excluded. Genes were assigned into several gene families (orthogroups) or recognised as species-specific genes (unique genes) according to the OrthoFinder results. The organisation and orientation of genes were visualised using R package 'gggenes'[89].

Phylogenetic analyses were performed in parallel to illustrate the identity and evolutionary pattern of genes of interest. Amino acid sequences were aligned to other orthologues from both metazoan and bacterial clades using MEGA v7.0[90]. Neighbour-joining phylogenetic trees were constructed using MEGA v7.0, where each phylogenetic node was analysed by bootstrapping with a value equal to 1000. A maximum-likelihood gene tree was constructed using IQ-TREE[91] with '-T AUTO -B 1000 -bnni --alrt 1000'. Gene trees were visualised using R package 'ggtree'[76].

**Differential tissue expression analyses and RT-qPCR**. Three adult female *T. tuberculata* were collected from culture and rinsed in double distilled water to remove surface contaminants. Animals were then dissected and various tissues (protocerebrum, nerve cord, vesicular gland, venom gland, fat body, Malpighian tubules, female dorsal gland, hindgut, midgut, leg and eggs collected from gravid females) were subject to total RNA extraction using TRIzol reagent following the manufacturer's instructions. Purified total RNA was dissolved in nuclease-free water. The synthesis of cDNA was carried out using the iScript™ gDNA Clear cDNA Synthesis Kit (BioRad, Catalogue number: 1725035BUN), following the enclosed instructions provided by the manufacturer. The synthesized cDNA was used as templates in quantitative real-time PCR. The amplification protocol performed: initial denaturation at 95 °C for 30 s, followed by 40 cycles of denaturation at 95 °C for 15 s, Primer annealing at 55 °C for 15 s and Extension at 72 °C for 15 s. Replicates were performed for each biological sample. The expression of each gene was normalized to the internal housekeeping gene, 60S ribosomal protein (60S) gene, and the relative fold changes of each target gene were calculated using the ΔΔCt method. Primer information is listed in Supplementary Data 3.

**In silico sesquiterpenoid system gene analyses and phylogenetic analyses**. Gene family sequences were first retrieved from the sequenced myriapod genomes using the tBLASTn algorithm. The identity of each retrieved gene was subsequently

submitted to the NCBI online nr database for reciprocal BLAST using the BLASTx algorithm. For phylogenetic analyses of gene families, DNA sequences were first translated into amino acid sequences and aligned to other reference sequences using MEGA 7.0 software[90], and phylogenetic trees (neighbour-joining algorithm) were constructed using MEGA 7.0 software, with phylogenetic node support estimated by performing 1000 bootstrap replicates. Sequences that could not be aligned (reported by MEGA 7.0), or were severely truncated or too divergent, were removed from phylogenetic analyses.

**Sesquiterpenoid hormone measurement**. Ten individuals of *T. tuberculata* and *H. holstii*, washed with ddH$_2$O, were placed in stainless steel grinding jars with balls of Retsch MM400 (Retsch, EW-04182-09). The grinding jars were first cooled in liquid nitrogen for 30 min. The samples were then placed inside the jar and homogenized with a frequency of 20 Hz for 30 s[92]. The homogenate was then transferred to a 10 mL glass centrifuge tube immediately, which contains 1 mL of acetonitrile (ACN), 1 mL of 0.9% (w/v) sodium chloride solution and 10 ng of JH III-D3 as an internal standard control. The mixture was ultrasonicated for 1 min, and subsequently vortexed and extracted twice with 2 mL of hexane. The hexane phase, that locate on the upper layer, was carefully removed and transferred to a new glass tube. The tube was opened and then dried under nitrogen flow, until all hexane evaporated. The residue was dissolved in 1 mL of ACN. The measurement of FA and MF was performed using a liquid chromatography method coupled to electrospray tandem mass spectrometry (LC-MS/MS)[92,93].

**In vitro hormone treatment of myriapods**. Adult female *T. tuberculata* (5 for FA, 9 for MF) and *H. holstii* (11 for FA and 6 for MF) were collected from laboratory culture and rinsed in double distilled water to remove surface contaminants. The fat body of each individual animal was dissected and placed in the well of a cell culture plate, with half of the tissue serving as a control and the other half serving as the experimental replicate. The well was filled with 198ul of Schneider *Drosophila* medium (Gibco™, Catalogue number: 21720024) containing 10% (v/v) heat-inactivated foetal bovine serum (Gibco™, Catalogue number: 10270106) and 1:100 penicillin-streptomycin (Gibco™, Catalogue number: 15140122). FA (Echelon Biosciences, Inc, Catalogue number: S-0151-10MG) and MF (Echelon Biosciences, Inc, S-0153-10MG) dissolved in acetone (Lab-Scan Analytical Sciences, Code number: C2501) were added to each experimental replicate at a final concentration of $10^{-6}$ M and $10^{-8}$ M. The control replicate received the same amount of acetone. The treatment lasted for 4 h in the dark and RNA extractions (see Transcriptome Sequencing) were performed immediately afterwards.

**Reporting summary**. Further information on research design is available in the Nature Research Reporting Summary linked to this article.

## Data availability

The final assemblies were submitted to NCBI Assembly under accession numbers WWPM00000000 (*Glomeris maerens*), JAAFCF000000000 (*Helicorthomorpha holstii*), WWPL00000000 (*Anaulaciulus tonginus*), JAAIVG000000000 (*Niponia nodulosa*), JAAFCE000000000 (*Trigoniulus corallinus*), JAHWFP000000000 (*Lithobius niger*), JAHWFO000000000 (*Rhysida immarginata*) and JAFIDM000000000 (*Thereuonema tuberculata*) in NCBI. The raw reads generated in this study were deposited to the NCBI database under the BioProject accessions PRJNA598061 (*Glomeris maerens*), PRJNA564202 (*Helicorthomorpha holstii*), PRJNA598060 (*Anaulaciulus tonginus*), PRJNA606398 (*Niponia nodulosa*), PRJNA564195 (*Trigoniulus corallinus*), PRJNA738717 (*Lithobius niger*), PRJNA701115 (*Rhysida immarginata*) and PRJNA699399 (*Thereuonema tuberculata*). The genome annotation files were deposited in the Figshare (https://doi.org/10.6084/m9.figshare.15088722). The databases are available for download from the following websites: eggNOG http://eggnog5.embl.de/download/eggnog_5.0/, GO http://geneontology.org/, KEGG https://www.genome.jp/kegg/pathway.html, and KOG https://www.hsls.pitt.edu/obrc/index.php?page=URL1144075392. Source data are provided with this paper.

## Code availability

The scripts for carrying out analyses of this study were deposited in Zenodo: https://doi.org/10.5281/zenodo.5718734[79] and https://doi.org/10.5281/zenodo.6482625[94].

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

## Acknowledgements

This work was funded by Hong Kong Research Grant Council General Research Fund (14100919 and 14100420), Hong Kong Research Grant Council Collaborative Research Fund (C4015-20EF), The Chinese University of Hong Kong Direct Grant (4053433, 4053489 and 4053547), and Environment and Conservation Fund (2018-82) to J.H.L.H. W.L.S. was supported by Sir Edward Youde Memorial Fellowship for Postgraduate Research Students. W.L.S. was supported by PhD studentship provided by The Chinese University of Hong Kong. A.H. is supported by a Biotechnology and Biological Sciences Research Council (BBSRC) David Phillips Fellowship (BB/N020146/1). T.B. is supported by a studentship from the Biotechnology and Biological Sciences Research Council-funded South West Biosciences Doctoral Training Partnership (BB/M009122/1).

## Author contributions

J.H.L.H. and A.H. designed and coordinated the project. W.L.S. carried our animal collection, husbandry and extraction of DNA and RNA for sequencing. W.N. carried out the final genome assembly with the help from J.H. and T.S. W.N. carried out transcriptome assembly and gene model prediction. Y.X. carried out macrosyntenic analysis, enrichment analysis and genome comparison for gene gain and loss. T.B. carried out repeat analysis with A.H. W.L.S. and Z.Q. carried out homeobox gene and sesquiterpenoid gene annotation. W.L.S. and W.N. carried out horizontal gene transfer analysis, with the help from X.Y. W.L.S. carried out the in vitro hormone treatment, real-time qPCR and subsequent analysis. H.M. carried out hormone measurements with Z.K. J.D.G., K.F.L., W.G.B., S.S.T., Z.K., A.H. and J.H.L.H. contributed the discussion of project at different stages. J.H.L.H. and W.L.S. wrote the initial manuscript; all authors revised and contributed to the final version of the text.

## Competing interests

The authors declare no competing interests.
