## [Peer Review File · Nature Communications]

Myriapod genomes reveal ancestral horizontal gene transfer and hormonal gene loss in millipedesREVIEWER COMMENTS

Reviewer #1 (Remarks to the Author):

Myriapods are the most poorly represented class in the world of arthropod genomics. For years there was only one sequenced species. The Hui lab added two millipedes a couple of years ago, and now add six more myriapods. With a total of 9 sequenced genomes (still a tiny number compared with other arthropod classes), they can now make genome-wide comparisons within the class and outside it and start drawing some evolutionary conclusions. The manuscript discusses a few specific points, and the supplementary data provides information for many additional comparisons in the future.

The main points the authors discuss are the evolution of genome size within myriapods, gene gain through horizontal gene transfer from bacteria, and one interesting case of gene loss in millipedes. They also discuss macro- and micro-synteny, with mostly expected results, and with an emphasis on the old favourite - the Hox cluster (no surprises there either).

The Hui group is experienced in work of this kind, and everything is done to expected standards, using the most current tools and analyses. I have no comments or criticism of the work itself. I have a few minor comments about some points detailed below, but nothing that requires more than minor edits. The question with this type of paper in the current genomic age is always, what is the correct publication platform? Is this paper "significant" enough to justify publication in Nature Communications? I think the answer is "yes". Although there are no major findings or breakthroughs, the fact that this work substantially increases our knowledge of an understudied group, and provides a body of data that can be useful for downstream work by other labs, coupled with the fact that the work is done to a high standard, shifts the balance in favour of publication.

Minor comments:

p. 3 l. 9 - I can't be accused of not appreciating myriapods, but I think that calling them "One of the most successful taxa of terrestrial animals" is a gross exaggeration. In fact, as a Class-level taxon, they are probably one of the least successful terrestrial classes, both in terms of species number and in terms of ecological diversity. Sorry. A similar statement appears later.

Genome size evolution - a quick check in the genome size database (genomesize.com) shows that small genome sizes are probably typical for geophilomorphs, whereas the other orders have much larger genomes. It's worth citing these data.

p. 8 l. 5 - Pleurostigmomorphs is misspelled.

p. 18. l. 16 - "mollusc genomes"? I suspect the authors copied a methods sentence from a different source and forgot to change the taxon name. If this is not a typo, please explain why molluscs are relevant here?

For some reason, the file I received did not have figure legends for the main figures.

Reviewer #2 (Remarks to the Author):

I enjoyed reading the manuscript titled "Myriapod genomes reveal ancestral horizontal gene transfer and hormonal gene loss in millipedes" by Wai Lok So and collaborators. The manuscript presents new genomes for myriapods, that together with other data are analysed to infer the evolution in myriapods of genome size, the transposable element (TE) content, genomic architecture (e.g., organisation of Hox genes), gene gains (horizontal gene transfer, HGT), gene losses (e.g., sesquiterpenoid hormone system), together with experimental physiological data. Overall, the analyses are robust and adhere to the standards of the field, displaying a large diversity of methods (from bioinformatics to physiology) and interesting findings (e.g., HGT). The article is well-written and presented. I'd like to congratulate the authors for their work and efforts.

My main criticism is more of style than content. The paper narrative is not rooted in biological questions and/or hypotheses, which are actually there but not articulated. For example, the abstract opens with a sentence on how diverse all animals are, followed by the role of gene gains and losses, then directly jumps to say some myriapod genomes have been sequenced, without justifying why the

focus on myriapods. Similarly, the Introduction gives a general overview of myriapod diversity, but does not introduce the very interesting topics and knowledge gaps that are indeed explored in the Results (TE, Hox cluster, HGT...) and jumps directly to the genome sequencing. Some of the controversies are directly introduced in the Results and Discussion. As a result, the paper is very dry and descriptive, and the reader is left wondering "why did they do this?".

On a technical level, I was surprised to see Neighbour Joining trees, which were superseded by probabilistic approaches long ago. These are not to the standards of the field and should be replaced by more suitable analyses like Maximum Likelihood trees in IQTree, which are almost equally fast and several orders of magnitude more accurate and reliable. Similarly, the taxon sampling in the comparative genomics analyses is very limited, especially for outgroups for which there is already a wealth of genomes available. This makes the inference of gains and losses in the early branches of the trees unreliable. A more diverse and representative outgroup selection (e.g., nematodes, chelicerates, etc.), would be needed to support the claims made about early myriapod genome evolution. I have some specific comments I hope the authors can address:

- 1) Page 2, line 31: The "and" should be replaced by a comma (there is already another "and" in this sentence in the next line).
- 2) Page 4, line 1: I am not sure "to de novo assembly" is a real verb, I'd replace it by "we generated a de novo assembly for...".
- 3) Page 4, line 8: I respectfully disagree with the BUSCO scores being high. I'd just report the values and not assess if they are high or low.
- 4) Page 5, line 11: similarly, "close to chromosome-level" is vague. I'd either be specific with the numbers/values or just remove the sentence.
- 5) Figure 2 is barely mentioned or discussed in the text; I'd move it to Supp data. There is no justification for the choice of *Thereuonema* for the comparisons in this figure.
- 6) Keeping with figures, it is not clear why the contents of Figure 3A were not split into two panels. It is very hard to follow to which subpanel the text is referring. I think that on page 5 line 10, when the text refers to Figure 3B it really means Figure 3A bis.
- 7) Page 5, line 2: I am not an expert, but I was wondering if the TE family content of different species can be compared to reconstruct when the TE expansions happened during myriapod evolution.
- 8) Page 5, line 19: while I do find the link between TEs and HGT fascinating, I think the text should mention that most literature suggests that the main driver for TE spread in genomes is population size (e.g., Lynch papers).
- 9) Page 6, line 8: the numbers of predicted genes look quite high. Can the text elaborate if these genes have been confirmed with other evidence, like transcriptome sequencing?
- 10) Page 7, line 9-10: there is no description of the Hox cluster status in other species of myriapods.
- 11) Page 8, line 7-8: I was wondering if these bacterial HGT genes have introns?
- 12) Page 13, line 3: typo in "artificial"
- 13) Page 14, line 11: reference missing in "<ref>".

Reviewer #3 (Remarks to the Author):

Review of "Myriapod genomes reveal ancestral horizontal gene transfer and hormonal gene loss in millipedes" by first authors So, Nong and Xie, and colleagues.

Overall this is the first excellent comparative genomic analysis of the Myriapoda and I believe it deserves to be accepted for publication with minor optional edits as described below.

General notes:

1. This is an important increase in the genomic data available for an important and underrepresented group – the myriapods and deserves publication in a high-profile journal for that alone.
2. The authors have gone beyond the standard single genome analysis and provided a comparative

genomics analysis of a large group – this is to be highly commended and adds significant intellectual interest.

3. The analysis of genome size variation and factors contributing to that size variation is excellent and extremely clear.

4. The 279 gene family gains for centipedes vs 16 for millipedes is a very interesting result. The finding of the enrichment in “enriched in the centipede ancestor, including arginine and proline metabolism, cholesterol metabolism, isoquinoline alkaloid biosynthesis, PI3K-Akt signaling, and serotonergic synapse.” Is an excellent observation and a powerful introduction to the phenotypic and genomic changes required for the shift to carnivorism.

5. The paragraph on homeobox genes is nice. “with subsequent loss of functional constraints of Hox3 in certain myriapod lineages only” might also be due to technical problems with the earlier assemblies. I’m not sure how best to indicate the possibility to the reader in as few words as possible. Something like: “with subsequent loss of functional constraints of Hox3 (or possible technical assembly problems) in certain myriapod lineages only”

6. The HGT and loss of gene content analyses are major accomplishments that deserve rapid publication.

Genome quality.

The genome qualities are generally pretty good. With the very latest technology they could be better, but you sequence with the technology you have at the time of funding, and I see no reason why the assemblies would dramatically change the conclusions or analysis of the work.

Minor text issues:

A biggish one is your data availability paragraph which reads:

“Data Availability The final chromosome assembly was submitted to NCBI Assembly under accession number JAFIDM000000000 in NCBI. The raw reads generated in this study have been deposited to the NCBI database under the BioProject accessions: PRJNA699399, the genome annotation files were deposited in the Figshare (<https://doi.org/10.6084/m9.figshare.15088722>).”

Of course you have more than one genome and you have assembly accession numbers in Fig 1 Table H.

Can you update the Data availability paragraph so that it points the reader too all the data you generated?

Line 23, I would say described species in “highly successful group of terrestrial arthropods containing ~16,000 species” unless you are specifically referencing an estimate of described and undescribed.

P 6 line 24-26: “These results suggest that divergent pathways were rapidly taken by centipede and millipede lineages following their evolution from the myriapod ancestor, which were shaped by differing ecological pressures.” It is not clear to me that this sentence adds much – unless you are emphasizing the “rapidly”?, and I think that optionally you could remove the sentence as I think it somewhat dilutes the rest of the paragraph. If you don’t it’s OK too.

Supplementary information:

This file is labeled So et al on p1, but as the first three authors are labelled as contributed equally it should be “So, Nong, and Xie et al”

The supplementary data is otherwise extensive and excellent.

Reviewer #1 (Remarks to the Author):

1) Myriapods are the most poorly represented class in the world of arthropod genomics. For years there was only one sequenced species. The Hui lab added two millipedes a couple of years ago, and now add six more myriapods. With a total of 9 sequenced genomes (still a tiny number compared with other arthropod classes), they can now make genome-wide comparisons within the class and outside it and start drawing some evolutionary conclusions. The manuscript discusses a few specific points, and the supplementary data provides information for many additional comparisons in the future.

The main points the authors discuss are the evolution of genome size within myriapods, gene gain through horizontal gene transfer from bacteria, and one interesting case of gene loss in millipedes. They also discuss macro- and micro-synteny, with mostly expected results, and with an emphasis on the old favourite - the Hox cluster (no surprises there either).

The Hui group is experienced in work of this kind, and everything is done to expected standards, using the most current tools and analyses. I have no comments or criticism of the work itself. I have a few minor comments about some points detailed below, but nothing that requires more than minor edits.

The question with this type of paper in the current genomic age is always, what is the correct publication platform? Is this paper "significant" enough to justify publication in Nature Communications? I think the answer is "yes". Although there are no major findings or breakthroughs, the fact that this work substantially increases our knowledge of an understudied group, and provides a body of data that can be useful for downstream work by other labs, coupled with the fact that the work is done to a high standard, shifts the balance in favour of publication.

>>>Response 1

We thank the reviewer for their complementary comments, and their positive and constructive review. We are very grateful for their time and input to help improve our manuscript.

2) Minor comments:

p. 3 l. 9 - I can't be accused of not appreciating myriapods, but I think that calling them "One of the most successful taxa of terrestrial animals" is a gross exaggeration. In fact, as a Class-level taxon, they are probably one of the least successful terrestrial classes, both in terms of species number and in terms of ecological diversity. Sorry. A similar statement appears later.

>>>Response 2

We thank the reviewer for this valid point, and agree that we were over-enthusiastic regarding the success of the myriapods! We have modified our statement now, as follows, referring to them simply as diverse:

-Page 2, line 21-23 – “Myriapoda, including centipedes and millipedes, represents a diverse group of terrestrial arthropods containing ~16,000 described species, that play important ecological roles in soil and forest ecosystems.”

-The similar statement on page 3 has now been deleted.

-Page 11, line 28-29 – “Our findings expand current understanding of the processes mediating genome evolution and macroevolutionary dynamics for Myriapoda, a diverse, abundant, and geographically widespread, but relatively neglected group of terrestrial arthropods.”

3) Genome size evolution - a quick check in the genome size database (genomesize.com) shows that small genome sizes are probably typical for geophilomorphs, whereas the other orders have much larger genomes. It's worth citing these data.

>>>Response 3

We thank the reviewer for pointing this out. The recommended data are now cited as suggested:

Page 12, line 7-10 – “Among myriapods, genome size was found to be highly variable. Centipedes generally have much larger genome sizes than millipedes, except for the geophilomorph centipede lineage (i.e. *Strigamia*), which had a comparatively small genome, consistent with other reports of small genome size for this group (Gregory 2022).”

4) p. 8 l. 5 - Pleurostigmomorphs is misspelled.

>>>Response 4

Many thanks for the correction. This spelling has been corrected (Page 9, line 13).

5) p. 18. l. 16 - "mollusc genomes"? I suspect the authors copied a methods sentence from a different source and forgot to change the taxon name. If this is not a typo, please explain why molluscs are relevant here?

>>>Response 5

We apologise for this typo! This issue has been corrected.

6) For some reason, the file I received did not have figure legends for the main figures.

>>>Response 6

We apologise for this oversight. The figure legends are now included on Page 23.

Reviewer #2 (Remarks to the Author):

7) I enjoyed reading the manuscript titled “Myriapod genomes reveal ancestral horizontal gene transfer and hormonal gene loss in millipedes” by Wai Lok So and collaborators. The manuscript presents new genomes for myriapods, that together with other data are analysed to infer the evolution in myriapods of genome size, the transposable element (TE) content, genomic architecture (e.g., organisation of Hox genes), gene gains (horizontal gene transfer, HGT), gene losses (e.g., sesquiterpenoid hormone system), together with experimental physiological data. Overall, the analyses are robust and adhere to the standards of the field, displaying a large diversity of methods (from bioinformatics to physiology) and interesting findings (e.g., HGT). The article is well-written and presented. I’d like to congratulate the authors for their work and efforts.

>>>Response 7

We thank the reviewer for their kind comments, and are very grateful for their time and efforts in help us to improve our manuscript.

8) My main criticism is more of style than content. The paper narrative is not rooted in biological questions and/or hypotheses, which are actually there but not articulated. For example, the abstract opens with a sentence on how diverse all animals are, followed by the role of gene gains and losses, then directly jumps to say some myriapod genomes have been sequenced, without justifying why the focus on myriapods. Similarly, the Introduction gives a general overview of myriapod diversity, but does not introduce the very interesting topics and knowledge gaps that are indeed explored in the Results (TE, Hox cluster, HGT...) and jumps directly to the genome sequencing. Some of the controversies are directly introduced in the Results and Discussion. As a result, the paper is very dry and descriptive, and the reader is left wondering “why did they do this?”.

>>>Response 8

We thank the reviewer for this point. We agree that the manuscript would benefit from a more explicit justification of the rationale for the study. We have now added a new paragraph justifying the phylogenetic relevance of a focus on myriapods (Page 3, lines 21-31), and added more detail to the subsequent paragraph (Page 4, lines 5-23). The additional text introduces the themes of genome size, gene family evolution, and developmentally relevant genes, to provide context surrounding their importance in the study of myriapod genomics and animal evolution more generally.

- Page 3, lines 21-28 – “In addition to their ecological relevance, myriapods occupy an important phylogenetic position in animal evolution, with analyses placing them as the extant sister group to Pancrustacea, the major animal clade containing insects and crustacean lineages (Giribet and Edgecombe 2019). Thus, consideration of the putative genomic character of the myriapod ancestor, and the subsequent divergent evolutionary pathways taken by centipede and millipede lineages, is of direct relevance to the study of arthropod evolutionary genomics. Yet, the current genomic resources available for myriapods are highly limited compared to other major arthropod lineages (Feron and Waterhouse 2022).”

- Page 4, lines 1-19 – “Considerable variation in genome size is a distinctive feature of metazoan genomes, but the drivers of observed differences remain poorly understood (Blommaert 2020). Consequently, we start by investigating genome size across Myriapoda, to examine the extent of variation and potential causative factors. We then consider genomic

features with potential roles in shaping the divergent evolutionary trajectories of centipedes and millipedes, towards active predation and detritivory respectively. Specifically, we examine the extent to which patterns in gene family evolution (gene gain, loss and horizontal transfer) underlie key ecological and morphological differences. Such analyses offer fundamental insights into the mode of animal evolution (Fernández & Gabaldón 2020), and provide an opportunity to assess whether genes that arose in other branches of life (such as fungi and bacteria), have driven the evolution of novel adaptations in animals following horizontal transfer (Husnik & McCutcheon 2018; Wybouw et al 2016). We also compare the content and organisation of developmentally relevant genes and pathways between centipedes and millipedes, to examine their respective roles in myriapod evolution. In particular, we consider homeobox genes, the content and organisation of which can have profound impacts on animal phenotypes (Holland 2012), and hormonal pathways which regulate arthropod development (Cheong et al 2015). Collectively, our findings significantly expand current knowledge of the myriapod genomics, and provide novel insights into the evolution of genome size, gene repertoire, and genetic pathways across myriapod diversity.”

9) On a technical level, I was surprised to see Neighbour Joining trees, which were superseded by probabilistic approaches long ago. These are not to the standards of the field and should be replaced by more suitable analyses like Maximum Likelihood trees in IQTree, which are almost equally fast and several orders of magnitude more accurate and reliable.

>>>Response 9

We thank the reviewer for the suggestion, and agree with this point. Maximum-likelihood trees are now estimated and included in the Supplementary Information. All gene assignments remain the same.

10) Similarly, the taxon sampling in the comparative genomics analyses is very limited, especially for outgroups for which there is already a wealth of genomes available. This makes the inference of gains and losses in the early branches of the trees unreliable. A more diverse and representative outgroup selection (e.g., nematodes, chelicerates, etc.), would be needed to support the claims made about early myriapod genome evolution.

>>>Response 10

This is a good suggestion, and we have taken this comment onboard. We have now re-tested our claims by reconstructing the ancestral situation and inference of gene gains and losses with the addition of diverse representative outgroups, including genomes of the scorpion *Centruroides sculpturatus*, spider *Stegodyphus mimosarum*, tick *Ixodes scapularis*, shrimp *Litopenaeus vannamei*, crab *Eriocheir sinensis*, beetle *Tribolium castaneum*, fly *Drosophila melanogaster*, tardigrade *Hypsibius dujardini*, nematode *Caenorhabditis elegans*, snail *Biomphalaria straminea*, oyster *Magallana hongkongensis*, human *Homo sapiens* and amphioxus *Branchiostoma floridae*.

The resultant inferred gene gains and losses are now included in the Supplementary Information. We report that all our claims regarding isoquinoline alkaloid biosynthesis remain robust and statistically significant. In addition, we have also found more centipede-specific gains as a result of the new analyses. This new information is now included in the main text, as detailed below:

Page 7, line 29-32 – Page 7, line 24-31 – “As shown in Figure 3C and the Supplementary Information, seven KEGG pathways were enriched in the centipede ancestor, including the adipocytokine signalling pathway, complement and coagulation cascades, isoquinoline alkaloid biosynthesis pathway, PPAR signalling pathway, SNARE interactions in vesicular transport, fatty acid biosynthesis and tyrosine metabolism. Considering that the enrichment of orthologues in several of these pathways involves lipid metabolism, immunity, and signal transduction, these may also have contributed to sensory and locomotory adaptations that facilitated the ecological shift to predation in centipedes.”

11) I have some specific comments I hope the authors can address: 1) Page 2, line 31: The “and” should be replaced by a comma (there is already another “and” in this sentence in the next line).

>>>Response 11

The word is now replaced by a comma as suggested.

12) 2) Page 4, line 1: I am not sure “to de novo assembly” is a real verb, I’d replace it by “we generated a de novo assembly for...”.

>>>Response 12

Page 4, line 27-28 - The sentence has now been changed to “we generated *de novo* genome assemblies for six myriapod species”

13) 3) Page 4, line 8: I respectfully disagree with the BUSCO scores being high. I’d just report the values and not assess if they are high or low.

>>>Response 13

Page 4, line 33 – Page 5, line 1 - This statement has now been modified to: “The assembled myriapod genomes possess BUSCO scores of 63.4-93.8% (mean = 82.5%).”

14) 4) Page 5, line 11: similarly, “close to chromosome-level” is vague. I’d either be specific with the numbers/values or just remove the sentence.

>>>Response 14

This sentence has now been removed.

15) 5) Figure 2 is barely mentioned or discussed in the text; I’d move it to Supp data. There is no justification for the choice of *Thereuonema* for the comparisons in this figure.

>>>Response 15

We agree with the reviewer, and have now moved the original Figure 2 to supplementary information (Supplementary Figure 1-3). The reason for choosing the centipede *T. tuberculata*

and millipedes *H. holstii* and *T. corallinus* is because these genomes have high scaffold N50s and BUSCO values.

16) 6) Keeping with figures, it is not clear why the contents of Figure 3A were not split into two panels. It is very hard to follow to which subpanel the text is referring. I think that on page 5 line 10, when the text refers to Figure 3B it really means Figure 3A bis.

>>>Response 16

The figure has now been divided into two (Figure 2A & 2B) as suggested.

17) 7) Page 5, line 2: I am not an expert, but I was wondering if the TE family content of different species can be compared to reconstruct when the TE expansions happened during myriapod evolution.

>>>Response 17

We thank the reviewer for this valid point. In response, to dissect the pattern further we have added additional analyses, bar chart figures to the main text (Figure 2C-D), an element-by-element matrix to the Supplementary Information (Supplementary Figure 64), and accompanying discussion to the text (Page 6, line 1-13). The bar charts of unique vs shared repeat families clearly illustrate the overwhelming pattern that most transposable element families are unique to the taxon in which they are annotated. Thus, demonstrating that most expansions have occurred independently in each lineage. This pattern is perhaps unsurprising given the extremely great distances that separate currently sampled myriapod lineages, and highlights the need for further sampling to better understand transposable element dynamics within the group. We are unable to accurately estimate the specific timings of individual transposable element group expansions, but the included repeat landscape plots (Supplementary Figure 60) provide an indication of the relative timings of expansions for each species and each major transposable element type.

- Page 5-6, lines 29-8 – “To further examine the evidence for a decline in TE content in *S. maritima* versus independent gains in the other three centipede lineages, we calculated the number of shared vs unique TE families among sampled myriapod species. The results demonstrated that the overwhelming proportion of TE families are unique to individual species among currently sampled myriapod diversity (Figure 2C-D; Supplementary Figure 64). Thus, the data strongly support a hypothesis of parallel independent gains of TE content in *R. immarginata*, *L. niger*, and *T. tuberculata*, rather than a reduction in TE content in *S. maritima*. More generally, these findings support the independent gain of new TE families as the main factor in shaping TE diversity across major divisions of myriapod diversity (Figure 2A). This pattern is perhaps unsurprising considering the ancient divergence dates estimated to separate sampled major myriapod lineages (e.g. estimated date of divergence between *Glomeris* and *Helicorhombomorpha* is 456 MYA, Fernández et al 2016, Kumar et al 2017).”

18) Page 5, line 19: while I do find the link between TEs and HGT fascinating, I think the text should mention that most literature suggests that the main driver for TE spread in genomes is population size (e.g., Lynch papers).

>>>Response 18

We thank the reviewer for raising this point. They are quite right that we should have included a caveat stating that other factors may also underlie the observed pattern, which we have now done, as follows

- Page 6, line 27-33 – Page 7, line 1-3 - “However, the correlates of TE variation remain poorly elucidated in arthropods and more generally, and we acknowledge that other life history traits may underlie patterns of TE abundance and diversity in myriapods. Regarding population size specifically, Lynch’s work has been quite controversial, and the negative correlation suggested between effective population size and genome size disappeared when shared ancestry was taken into account, using comparative phylogenetic approaches (e.g. Whitney & Garland 2010). A discussion of some of the complexities is provided by Whitney et al (2011), which points out that among other issues, population size is often highly correlated with many other life history traits, and its relative importance compared to these other factors remains far from clear.”

19) Page 6, line 8: the numbers of predicted genes look quite high. Can the text elaborate if these genes have been confirmed with other evidence, like transcriptome sequencing?

>>>Response 19

Yes - we used the protein uniprot_sprot database and transcriptomic data as evidence for gene annotation. The information has been included in Supplementary Data file 1. The following information is now included in the main text:

Page 7, line 14-15 – “The gene annotation was conducted using proteins from the uniprot_sprot database and transcriptomic data (Supplementary Data 1).”

20) Page 7, line 9-10: there is no description of the Hox cluster status in other species of myriapods.

>>>Response 20

The following information has now been added:

Page 8, line 12-13 – “In the other myriapod genomes investigated here, Hox genes are scattered across different scaffolds.”

21) Page 8, line 7-8: I was wondering if these bacterial HGT genes have introns?

>>>Response 21

Page 9, line 2-4 - Among the 65 genes within the 9 myriapod genomes, 32 of them do not have introns and 33 possess introns. This information has been added to the main text and Supplementary Data 8-18 for each HGT candidate.

22) Page 13, line 3: typo in “artificial”

>>>Response 22

Page 14, line 14 - This typo has been corrected.

23) Page 14, line 11: reference missing in “<ref>”.

>>>Response 23

The missing reference has now been added.

Reviewer #3 (Remarks to the Author):

24) Review of “Myriapod genomes reveal ancestral horizontal gene transfer and hormonal gene loss in millipedes” by first authors So, Nong and Xie, and colleagues.

Overall this is the first excellent comparative genomic analysis of the Myriapoda and I believe it deserves to be accepted for publication with minor optional edits as described below.

General notes:

1. This is an important increase in the genomic data available for an important and underrepresented group – the myriapods and deserves publication in a high-profile journal for that alone.

2. The authors have gone beyond the standard single genome analysis and provided a comparative genomics analysis of a large group – this is to be highly commended and adds significant intellectual interest.

3. The analysis of genome size variation and factors contributing to that size variation is excellent and extremely clear.

4. The 279 gene family gains for centipedes vs 16 for millipedes is a very interesting result. The finding of the enrichment in “enriched in the centipede ancestor, including arginine and proline metabolism, cholesterol metabolism, isoquinoline alkaloid biosynthesis, PI3K-Akt signaling, and serotonergic synapse.” Is an excellent observation and a powerful introduction to the phenotypic and genomic changes required for the shift to carnivorism.

5. The paragraph on homeobox genes is nice. “with subsequent loss of functional constraints of Hox3 in certain myriapod lineages only” might also be due to technical problems with the earlier assemblies. I’m not sure how best to indicate the possibility to the reader in as few words as possible. Something like: “with subsequent loss of functional constraints of Hox3 (or possible technical assembly problems) in certain myriapod lineages only”

6. The HGT and loss of gene content analyses are major accomplishments that deserve rapid publication.

>>>Response 24

We thank the reviewer for the kind appreciation and constructive comments.

For the paragraph on the homeobox genes, the relevant statement has been revised as suggested:

Page 8, line 15-16 – “with subsequent loss of functional constraints for *Hox3* (or possible technical assembly problems) in certain myriapod lineages only (Figure 3A).”

25) Genome quality.

The genome qualities are generally pretty good. With the very latest technology they could be better, but you sequence with the technology you have at the time of funding, and I see no reason why the assemblies would dramatically change the conclusions or analysis of the work.

>>>Response 25

We agree with the reviewer and thank them for their understanding.

26) Minor text issues:

A biggish one is your data availability paragraph which reads:

“Data Availability The final chromosome assembly was submitted to NCBI Assembly under accession number JAFIDM000000000 in NCBI. The raw reads generated in this study have been deposited to the NCBI database under the BioProject accessions: PRJNA699399, the genome annotation files were deposited in the Figshare (<https://doi.org/10.6084/m9.figshare.15088722>).”

Of course you have more than one genome and you have assembly accession numbers in Fig 1 Table H. Can you update the Data availability paragraph so that it points the reader too all the data you generated?

>>>Response 26

We thank the reviewer for pointing out this issue. The Data Availability paragraph has now been amended as follows:

Page 21, line 2-14 - “The final assemblies were submitted to NCBI Assembly under the following accession numbers: WWPM000000000 (*Glomeris maerens*), JAAFCE000000000 (*Helicorthomorpha holstii*), WWPL000000000 (*Anaulaciulus tonginus*), JAAIVG000000000 (*Niponia nodulosa*), JAAFCE000000000 (*Trigoniulus corallinus*), JAHWFP000000000 (*Lithobius niger*), JAHWFO000000000 (*Rhysida immarginata*), and JAFIDM000000000 (*Thereuonema tuberculata*) in NCBI. The raw reads generated in this study were deposited to the NCBI database under the BioProject accessions: PRJNA598061 (*Glomeris maerens*), PRJNA564202 (*Helicorthomorpha holstii*), PRJNA598061 (*Anaulaciulus tonginus*), PRJNA606398 (*Niponia nodulosa*), PRJNA564195 (*Trigoniulus corallinus*), PRJNA738717 (*Lithobius niger*), PRJNA701115 (*Rhysida immarginata*), and PRJNA699399 (*Thereuonema tuberculata*). The gene annotation files were deposited in Figshare: <https://doi.org/10.6084/m9.figshare.15088722>. The scripts for carrying out analyses of this study were deposited in github: <https://github.com/xieyichun50/Myriapod-genomes>.”

27) Line 23, I would say described species in “highly successful group of terrestrial arthropods containing ~16,000 species” unless you are specifically referencing an estimate of described and undescribed.

>>>Response 27

Regarding the description of myriapods, this has now been changed as suggested:

Page 2, line 21-22 – “Myriapoda, including centipedes and millipedes, represents a diverse group of terrestrial arthropods containing ~16,000 described species, that play important and diverse ecological roles in soil and forest ecosystems.”

28) P 6 line 24-26: “These results suggest that divergent pathways were rapidly taken by centipede and millipede lineages following their evolution from the myriapod ancestor,

which were shaped by differing ecological pressures.” It is not clear to me that this sentence adds much – unless you are emphasizing the “rapidly”?, and I think that optionally you could remove the sentence as I think it somewhat dilutes the rest of the paragraph. If you don’t it’s OK too.

>>>Response 28

The sentence has now been removed as suggested.

29) Supplementary information:

This file is labeled So et al on p1, but as the first three authors are labelled as contributed equally it should be “So, Nong, and Xie et al”

The supplementary data is otherwise extensive and excellent.

>>>Response 29

The file has now been re-labelled as suggested.

REVIEWERS' COMMENTS

Reviewer #2 (Remarks to the Author):

I reread with pleasure this manuscript. I would like to thank the efforts made by the authors to improve the paper, they have addressed my concerns.